



# Remote sensing of methane point sources with the MethaneAIR airborne spectrometer

Luis Guanter[1,2], Jack Warren[3], Mark Omara[3], Apisada Chulakadabba[4,5], Javier Roger[2], Maryann Sargent[5], Jonathan E. Franklin[5], Steven C. Wofsy[5], and Ritesh Gautam[3]

[1]Environmental Defense Fund, Amsterdam, The Netherlands
[2]Research Institute of Water and Environmental Engineering (IIAMA), Universitat Politècnica de València, València, Spain
[3]Environmental Defense Fund, New York, New York, USA
[4]Environmental Sensing and Modeling, Technical University of Munich, Munich, Germany
[5]Harvard John A. Paulson School of Engineering and Applied Sciences, Harvard University, Cambridge, MA, USA

**Correspondence:** Luis Guanter
(lguanter@edf.org)

**Abstract.**

The MethaneAIR imaging spectrometer was originally developed as an airborne demonstrator of the MethaneSAT satellite mission. MethaneAIR enables accurate methane concentration retrievals from high spectral resolution measurements in the 1650-nm methane absorption feature at a nominal spatial sampling of $5 \times 25$ m. In this work, we present a computationally-
efficient data processing chain optimized for the detection and quantification of methane plumes with MethaneAIR. It involves the retrieval of methane concentration enhancements ($\Delta XCH_4$) with the high-precision matched-filter retrieval, which is applied to 1650-nm retrievals for the first time. Methane plumes are detected through visual inspection of the resulting $\Delta XCH_4$ maps. We evaluated the performance of this processing scheme with simulated plumes, intercomparison with other methods, and controlled methane releases. We applied this processing chain to MethaneAIR data mosaics acquired over the Permian
Basin during flights in 2021 and 2023, which resulted in the detection of hundreds of point sources above 100-200 kg/h, with a conservative detection limit around 120 kg/h. Our results show the consistency of MethaneAIR's $\Delta XCH_4$ matched-filter retrievals, and their potential for the detection and quantification of methane point sources across large areas.

## 1 Introduction

The remote detection and quantification of methane point sources is crucial to guide methane emission mitigation efforts.
Airborne and spaceborne imaging spectrometers are being widely used for this application. Optical imaging spectrometers record the light reflected by the Earth surface after interaction with the atmosphere in hundreds of contiguous spectral channels. These spectrally-resolved measurements allow the quantification of atmospheric methane concentrations from the 1650 or 2300 nm shortwave infrared (SWIR) spectral regions in which methane absorbs radiation. The resulting methane concentration maps can be used to identify and quantify methane plumes, which can be attributed to the corresponding sources.

We can classify the imaging spectrometers with potential for methane mapping into two different instrument classes, defined by the instrument's spectral configuration. First, we have the spectrometers sampling the entire solar spectrum ($\sim$400–2500 nm)





with a relatively coarse spectral sampling between 5 and 10 nm, and a relatively high spatial resolution (a few meters in the case of some airborne instruments). Methane retrievals for this type of instrument exploit the 2300 nm methane feature. Most of the developments towards the detection and quantification of methane point sources are based on previous work with the AVIRIS

and AVIRIS-NG airborne spectrometers, which belong to this instrument class. For example, Roberts et al. (2010) detected methane emissions from a marine geological seep source with AVIRIS; Thorpe et al. (2014) and Thorpe et al. (2017) discussed methane retrieval methods for AVIRIS and AVIRIS-NG; Frankenberg et al. (2016) used AVIRIS-NG to survey methane point souces in the Four Corners region (USA); and Cusworth et al. (2022) assessed the methane emissions from different U.S. basins with AVIRIS-NG.

The second group of methane-sensitive spectrometers sample a narrow spectral window around the 1650 nm methane absorption, with a sub-nanometer spectral sampling. The GHGSat instruments (spaceborne and airborne) and the Methane Airborne MAPper (MAMAP) and MAMAP-2D airborne spectrometers belong to this category. The 1-D (profiler) version of the MAMAP spectrometer has been operating since the 2010s (Krings et al., 2011). For example, MAMAP was used to map methane emissions in the Upper Silesian Coal Basin in southern Poland (Krautwurst et al., 2021). A 2-D configuration (imager)

of the instrument is now available (Gerilowski et al., 2011). In general, the instruments sampling a narrow spectral window around the 1650 nm absorption with a high spectral resolution can better disentangle the methane signal from that of surface structures. This makes these instruments to be less affected by surface-driven systematic retrieval errors, which usually comes at the expense of a higher retrieval noise.

The MethaneAIR instrument belongs to the spectrometer class sampling the 1650 nm window. It was developed as the

airborne demonstrator of the MethaneSAT satellite mission, launched on 4 March 2024 (Environmental Defense Fund, 2021). Unlike other airborne imaging spectrometers solely used for point sources, MethaneAIR is intended to provide information on both high-emitting point sources and area sources, and subsequently on total regional emissions. To achieve its primary goals of total regional emission quantification, MethaneAIR is designed to fly from high-altitudes (typically 40,000 ft above ground). This allows to map wider areas faster while disaggregating emissions from area and point sources, at the expense of some

loss in spatial resolution compared to airborne systems flying at lower altitudes. In addition, the need to sample area sources motivates that an accurate methane concentration ($XCH_4$) retrieval based on the $CO_2$-proxy method (Chan Miller et al., 2024) is implemented in MethaneAIR's operational processing chain. The good performance of MethaneAIR's $CO_2$-proxy $XCH_4$ retrieval for the quantification of methane plumes is shown in Chulakadabba et al. (2023) and El Abbadi et al. (2024). However, this retrieval is computationally-demanding, and is not optimized for point sources, for which a high-precision retrieval would

be required to reduce the plume detection limits.

In this work, we delve into maximizing the effectiveness of MethaneAIR measurements to rapidly process data across large areas with goals of improving plume detection limits. We propose a data processing scheme optimized for the detection of methane plumes, namely through a high-precision data-driven methane concentration retrieval based on the matched-filter concept, and on the visual inspection of the resulting methane concentration maps. We tested this processing chain on large-scale

flight campaigns performed with MethaneAIR over the Permian Basin (USA) as well as over a controlled-release experiment in Arizona (USA) in recent years.



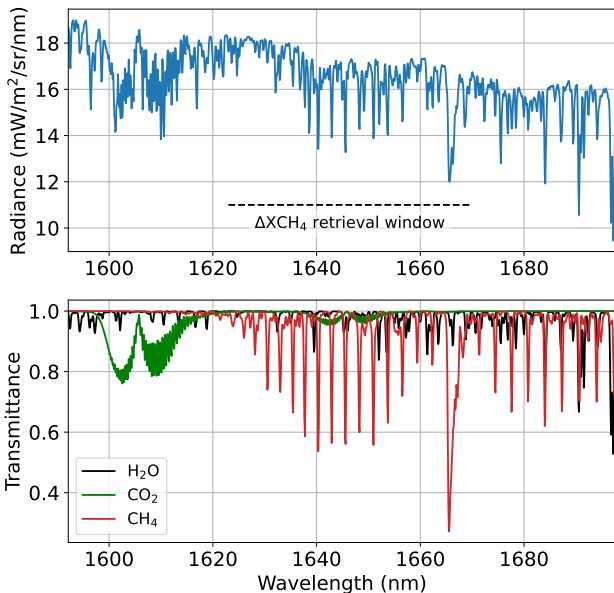

**Figure 1.** MethaneAIR's spectral coverage and sensitivity to atmospheric gases. A real MethaneAIR at-sensor radiance spectrum is shown in the top panel. The spectral window used for the retrieval of methane concentration enhancements ($\Delta$XCH$_4$) in this work is depicted with a dash line. Arbitrary spectral transmittance spectra for methane, CO$_2$ and water vapor convolved with MethaneAIR's spectral response functions are displayed in the bottom panel.

## 2    Materials and Methods

### 2.1    MethaneAIR's specifications and data products

An overview of the MethaneAIR instrument and a list of its technical specifications is provided in Staebell et al. (2021). MethaneAIR is typically flown at a 12 km altitude, which leads to a swath width of about 7.5 km, with an across-track pixel size of about 5 m and an along-track pixel of 25 m. MethaneAIR's methane band covers the 1592–1680 nm window, with a spectral resolution (full-width at half-maximum of the spectral response function) of about 0.3 nm, and a spectral sampling of 0.1 nm. As it is shown in Fig. 1, it samples the methane absorption feature around 1650 nm, and also the CO$_2$ absorption feature around 1610 nm, which is used for the CO$_2$-proxy methane retrieval (Chan Miller et al., 2024).

The conversion of MethaneAIR's raw level-0 data into level-1B spectral radiance data cubes is described in Conway et al. (2024). Subsequent processing levels in MethaneAIR's operational processing chain include dry column methane mixing ratio (XCH$_4$) maps in the original instrument coordinates, as the level-2 product (Chan Miller et al., 2024); geoprojected and orthorectified XCH$_4$ mosaics as the level-3 product, and information on methane fluxes (both detected plumes from high-emitting point sources and spatially-distributed area fluxes) as the level-4 product.



70  This study uses as input MethaneAIR level-1B data, which corresponds to calibrated and georeferenced radiance spectra. MethaneAIR's level-1B spectral radiance datasets are stored as "granules" of $301 \times 1280$ spatial pixels (along-track $\times$ across-track). The full flightline is reconstructed after appending all granules in the along-track direction. For the across-track direction, 1280 is the size of the detector's focal plane array, but only a fraction of it (typically, 863 pixels) is illuminated. When the data are spatially binned across-track (5 spatial pixels combined into 1) in order to generate lighter data files with

75 square pixels, the dimensions of the illuminated part of a single granules is $301 \times 172$ pixels ($7.5\,\mathrm{km}$ along track, $4.7\,\mathrm{km}$ across track, for nominal operations at $12\,\mathrm{km}$ altitude).

## 2.2 $\Delta$XCH$_4$ retrieval

A useful variable for the detection and quantification of methane point sources from remote sensing data is the per-pixel methane concentration enhancement ($\Delta$XCH$_4$). For the retrieval of $\Delta$XCH$_4$ maps with MethaneAIR, we have adapted the

80 matched-filter retrieval. This has been widely applied to a range of airborne and spaceborne spectrometers sampling the $2300\,\mathrm{nm}$ methane absorption with a 5-10 nm spectral resolution (e.g. Thompson et al., 2015, 2016; Foote et al., 2020; Cusworth et al., 2021; Irakulis-Loitxate et al., 2021; Guanter et al., 2021; Roger et al., 2024), but it has not been previously tested on MethaneAIR-like spectrometers measuring in the $1650\,\mathrm{nm}$ window with a $0.1\,\mathrm{nm}$ spectral sampling.

  The matched-filter retrieval expresses the input radiance spectra as the perturbation of an average radiance spectrum by a

85 change in the methane column concentration. This is modelled as a so-called target spectrum, which represents the radiative transfer signal of a unit methane absorption. Following the notation by Thompson et al. (2016), if we name $\Delta$XCH$_4$ as $\widehat{\alpha}$, the matched-filter takes the form

$$\widehat{\alpha}(\boldsymbol{x}) = \frac{(\boldsymbol{x}-\boldsymbol{\mu})^T \Sigma^{-1} \boldsymbol{t}}{\boldsymbol{t}^T \Sigma^{-1} \boldsymbol{t}}, \tag{1}$$

where $\boldsymbol{x}$ is the spectrum under analysis, $\boldsymbol{\mu}$ and $\Sigma$ are the mean and covariance of the background spectral radiance, and $\boldsymbol{t}$ is the

90 target spectrum representing the perturbation of the background radiance signal by a methane enhancement. The $\boldsymbol{t}$ spectrum has units of radiance over methane column concentration, and is generated as $\boldsymbol{\mu} \cdot \boldsymbol{k}$, with $\boldsymbol{k}$ being a unit methane absorption spectrum calculated using radiative transfer simulations.

  The variable $\boldsymbol{\mu}$ is calculated on a per-column basis in order to account for the different radiometric responses of detector elements across-track. In the case of the target spectrum, one single instance of $\boldsymbol{k}$ is generated at high spectral resolution

95 for the entire image considering the illumination and observation angles of the acquisition, but the spectral convolution with the MethaneAIR spectral response function is performed on a per-column basis in order to account for potential across-track variations of the instrument spectral response, as caused by e.g. changes in the thermal environment of the sensor. An initial step in our processing chain detects and corrects potential global spectral shifts in MethaneAIR spectral calibration.

  Regarding the inverse covariance matrix $\Sigma^{-1}$, it was calculated on a per-column basis in our first implementation of the

100 retrieval. However, we noted that the relatively low number of along-track samples (301) in the level-1B data granules (see Section 2.1) affected the calculation of $\Sigma^{-1}$ so that the retrieval was low-biased. This effect has also been found in the processing of short flightlines from the AVIRIS-NG sensor (Ayasse et al., 2023). To overcome this issue, we calculate a global $\Sigma^{-1}$





from all the pixels in the granule, which proved to solve the underestimation of $\Delta XCH_4$ while being effective to account for across-track offsets thanks to the per-column calculation of $\mu$. This granule-level $\Sigma^{-1}$ calculation is allowed by MethaneAIR's
uniform spectral response in the across-track direction (very low spectral smile effect).

The 1623–1670 nm window was selected for the matched-filter retrieval, as it provides a good compromise between the number of methane lines available for the retrieval and the potential disturbance by other gases (see Fig. 1). Other narrower fitting windows were tested, but they yielded higher precision errors without a clear gain in retrieval accuracy.

### 2.3 Plume detection and quantification

Methane plumes are detected through visual inspection of the $\Delta XCH_4$ maps generated from each level-1B granule, following the approach described in Guanter et al. (2021) for the PRISMA spaceborne spectrometer. In short, the candidate plumes identified through a first screening based on visual inspection are compared with the input spectral radiance data at the continuum of the 1650 nm absorption feature to discard false positives due to surface patterns. The resulting plume candidates are co-registered with very high resolution images of the area. The candidate plume is considered to be a true detection if it originates
from a point where potentially-emitting infrastructure is located according to the very high resolution image.

The relatively low sensitivity of MethaneAIR $\Delta XCH_4$ retrievals to the background surface would allow to implement an automatic detection process for the larger plumes using thresholds on $\Delta XCH_4$ or machine learning segmentation and classification methods (e.g. Joyce et al., 2023; Růžička et al., 2023). However, we opted for the manual approach in order to ensure that the maximum number of plumes were properly detected. This method also minimizes the occurrence of false positives.

For the estimation of emission rates ($Q$) from the detected plumes, we use the integrated mass enhancement (IME) approach (Frankenberg et al., 2016; Varon et al., 2018). Following the mass-balance principle, the total mass enhancement in the plume is related to the magnitude of the emission with a parameterisation dependent on wind speed, as

$$Q = \frac{U_{\text{eff}} \cdot \text{IME}}{L}. \tag{2}$$

This model calculates an IME in kg units as the total excess mass of methane contained in the plume. Plumes are manually
delineated in the $\Delta XCH_4$ maps using a Python script that has been implemented for this purpose. As proposed by Varon et al. (2018), we use an effective wind speed $U_{\text{eff}}$ in order to account for eddy-scale turbulence at MethaneAIR's spatial resolution, combined with the effects of retrieval noise. This $U_{\text{eff}}$ is related to the 10-m wind speed $U_{10}$ as

$$U_{\text{eff}} = 0.34 \cdot U_{10} + 0.42, \tag{3}$$

which was proposed by Maasakkers et al. (2022) for GHGSat. GHGSat and MethaneAIR share a similar spatial resolution
($\sim 25$ m) and a comparable retrieval noise (both instruments rely on high spectral resolution measurements in the 1650 nm window). $U_{10}$ data is taken from the GEOS-FP meteorological reanalysis product (GEOS-Chem, 2024). Errors in $Q$ estimates are obtained from the propagation of $\Delta XCH_4$ retrieval errors and a 50% uncertainty in wind speed through Eq. 2. The 50% uncertainty in wind speed is chosen as a conservative estimate for this variable, which drives the uncertainty of $Q$ estimations.

A more sophisticated implementation of the IME model for MethaneAIR, the modified IME (mIME) model, was proposed
by Chulakadabba et al. (2023). They assumed a logarithmic dependence between $U_{\text{eff}}$ and $U_{10}$. For $U_{10}$, they used the 10 m





root-mean-square wind obtained from each large-eddy simulation (LES) realization specifically run for the case of interest, rather than relying on operational meteorological products. However, we have chosen the more simple approach based on GEOS-FP winds because we have to run the $Q$ estimates for a large number of plumes.

We have intercompared our $Q$ estimates from the IME model with those from the mIME model, and also with the divergence
integral (DI) $Q$ estimation method. Both were developed for MethaneAIR and thoroughly validated with controlled-release tests (Chulakadabba et al., 2023). These two $Q$ estimation methods are more challenging to run over a large number of plumes than our basic IME method, but can provide an ideal reference to assess the performance of our IME-based $Q$ estimates.

### 2.4  End-to-end simulations of $\Delta$XCH$_4$ retrievals

We have used simulations to assess potential retrieval biases. We embedded simulated synthetic methane plumes into real
MethaneAIR level-1B data cubes. The simulated plumes were generated with the LES extension of the Weather Research and Forecasting model (WRF-LES). Concentrations in WRF-LES plumes were scaled to recreate a range of $Q$ values.

This mixed forward simulation approach combining real radiance data with simulated plumes has already been used for sensitivity analysis of high-resolution methane-sensitive instruments (Guanter et al., 2021; Roger et al., 2024; Gorroño et al., 2023). The use of real radiance data ensures that the actual measurement noise and potential radiometric and spectral offsets
are intrinsically included in the simulation.

### 2.5  MethaneAIR datasets used in this study

We evaluated MethaneAIR's potential for surveying methane point sources across large oil and gas basins using level-1B data from several MethaneAIR flight campaigns. In this work, we report results from the analysis of two MethaneAIR research flights focused on the Permian Basin (USA), where a high concentration of active methane sources can be found. Those
Permian Basin flights took place on 6 August 2021 ("RF06" flight) and on 20 July 2023 ("MX025" flight), and covered a region of about $120{\times}80\,\mathrm{km}^2$ including the Delaware sub-basin of the Permian Basin's oil and gas field with flights longer than 2 hours.

In addition, we processed data from another research flight, RF01E, which was carried out on 25 October 2022 over a single-blind volume-controlled methane-release experiment near Phoenix (USA) (Chulakadabba et al., 2023).

## 3  Results

### 3.1  $\Delta$XCH$_4$ retrieval performance

Results from the processing of a sample data granule of the RF06 campaign are displayed in Fig. 2, which shows a map of the input at-sensor radiance at 1623 nm (shortest wavelength in the retrieval window, see Fig.1), and the corresponding $\Delta$XCH$_4$ map. The processing involved $\Delta$XCH$_4$ retrieval, plume detection, and $Q$ estimation using the IME model. Four plumes were
detected through the visual inspection process, with $Q$ ranging from $87{\pm}33\,\mathrm{kg/h}$ to $512{\pm}180\,\mathrm{kg/h}$. It can be observed that





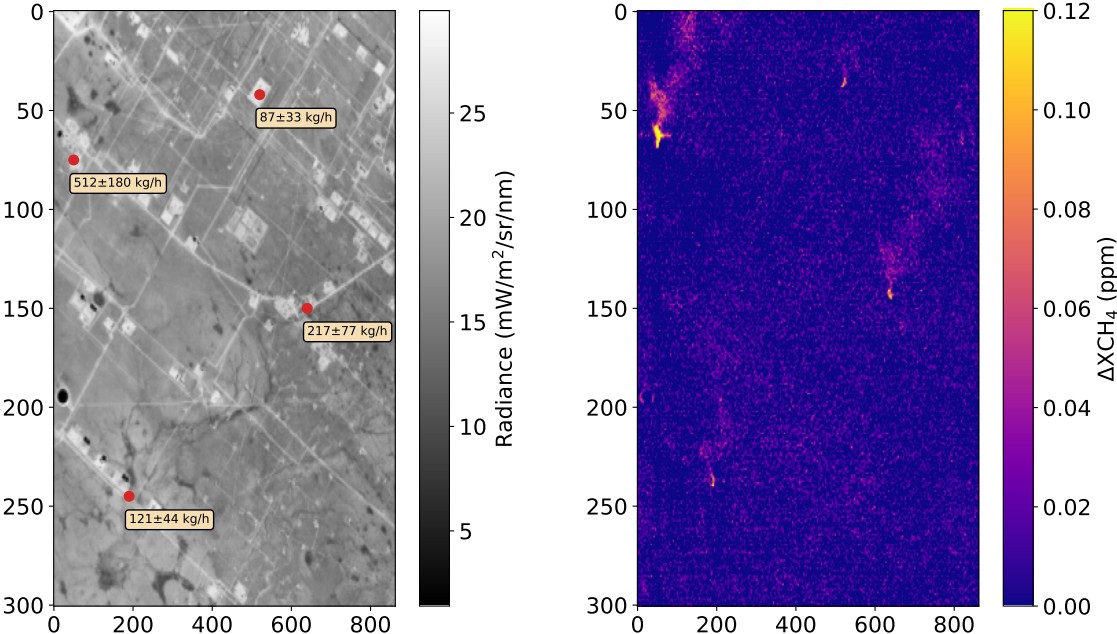

**Figure 2.** $\Delta$XCH$_4$ map retrieved from a MethaneAIR data granule from the RF06 Permian campaign. A map of the at-sensor radiance at 1623 nm is shown on the left panel, and the retrieved $\Delta$XCH$_4$ map is displayed on the right. The red points and the text boxes on the radiance map depict the location and flux rate of the four plumes detected in this subset.

these four plumes clearly stand above the background noise, although an automatic detection and segmentation of the smaller plumes would have been challenging. It can also be seen that there is a very low occurrence of systematic outliers in the $\Delta$XCH$_4$ maps despite the relatively high variability in the surface patterns, unlike the case of coarser spectral resolution instruments (Jongaramrungruang et al., 2021).

Further insights on the impact of the surface reflectance and spatial heterogeneity on the retrieval are provided in Fig. 3. It compares the intensity and spatial variability in at-sensor radiance with those of the retrieved $\Delta$XCH$_4$ for selected granules from the RF06 and RF01E flights where no methane plumes were detected. The spatial sampling is MethaneAIR's native 5×25 m. The results show that the $\Delta$XCH$_4$ variability is very close to a normal distribution, even for the RF01E granule for which the input radiance was far from Gaussian. The standard deviation is 33 and 38 parts-per-billion (ppb) for the RF06 and

RF01E granules, respectively. We interpret those numbers as the retrieval 1-$\sigma$ error for those granules. This 1-$\sigma$ error combines the per-pixel retrieval noise (measurement noise propagated to $\Delta$XCH$_4$ retrieval noise for each input spectrum), the variability introduced by the sensitivity of the retrieval to the surface spectral reflectance, and the potential contribution of methane sources in or close to the data granule under analysis. The lower 1-$\sigma$ error is found for the RF06 granule, which is consistent with the higher and more spatially-uniform at-sensor radiance. However, it must be remarked that the Permian Basin presents a high





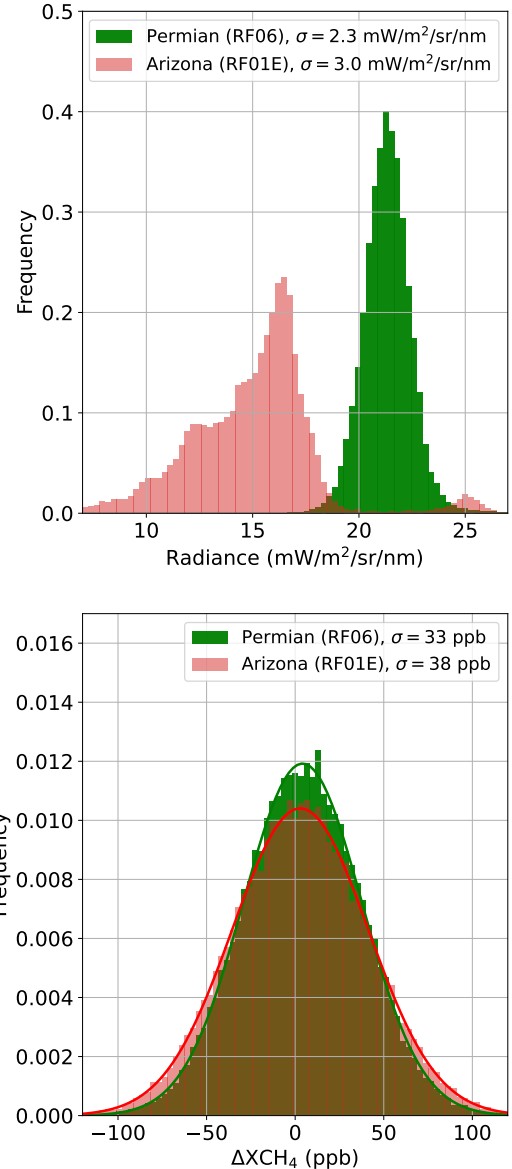

**Figure 3.** Variability in at-sensor radiance at 1623 nm (top) and retrieved $\Delta XCH_4$ (bottom) for sample subsets from the Permian Basin and Arizona campaigns (RF06 and RF01E, on 6 August 2021 and 25 October 2022, respectively).

concentration of methane point sources, so it is possible that part of the variability captured in the $\sigma$ calculated for the RF06 granule is due to methane plumes outside the analyzed granule or below MethaneAIR's detection limit.

A comparison between the matched-filter $\Delta XCH_4$ retrieval and the $CO_2$-proxy $XCH_4$ retrieval implemented in MethaneAIR's operational processing chain is shown in Fig. 4 for a subset of the granule displayed in Fig. 2. $\Delta XCH_4$ is calculated from the





XCH$_4$ generated by the CO$_2$-proxy through the removal of the XCH$_4$ background, which is estimated as a single offset from the plume-free pixels in the subset. The comparison of the two retrievals shows that the $\Delta$XCH$_4$ values from the data-driven matched-filter retrieval agree well with the more sophisticated CO$_2$-proxy XCH$_4$ retrieval, that has been thoroughly validated (Chan Miller et al., 2024). Also, we observe that the retrieval noise is lower for the matched-filter retrieval, namely $\sigma$ of 34 ppb for the CO$_2$-proxy retrieval and 23 ppb for the matched-filter, which enables the detection of a smaller plume on the right hand side of the matched-filter map. Note that these numbers are for a 25×25 m sampling, whereas the $\sigma$ values in Fig. 3 were for the native 5×25 m sampling. The higher retrieval precision error of the CO$_2$-proxy retrieval can be explained by the fact that the per-pixel normalization of the methane retrieval by the retrieved per-pixel CO$_2$ column density adds noise to the methane product. From this comparison, we conclude that the $\Delta$XCH$_4$ maps generated with the matched-filter retrieval can lead to lower plume detection limits than the CO$_2$-proxy retrieval because of their higher SNR, without an observable drop in retrieval accuracy. Nevertheless, total-column XCH$_4$ retrievals from the CO$_2$-proxy (as opposed to the $\Delta$XCH$_4$ retrievals by the matched-filter) are required for the estimation of area- and total-emission budgets, which is a key application of MethaneAIR. This implies that the matched-filter retrieval is not an alternative to the CO$_2$-proxy for the calculation of area and total methane fluxes with MethaneAIR.

We have further tested the consistency of the matched-filter $\Delta$XCH$_4$ retrievals by means of simulated plumes. A comparison between the input and the retrieved methane concentration enhancement from a simulated plume ($Q$=500 kg/h, $U_{10}$=3.4 m/s) is shown in Fig. 5. The plume was embedded into a real MethaneAIR granule following the procedure described in Sec. 2.4. There is a good agreement in the peak $\Delta$XCH$_4$ values between the simulated and the retrieved plume, which is evidenced by the lack of spatial structures in the difference map at the right-hand side of Fig. 5. On the other hand, the effect of retrieval noise is relatively large, causing that some of the lower methane concentration patches within the plume fall below the noise level. This needs to be considered when assessing potential error sources in the $Q$ estimation process. This issue is partly alleviated by the IME/$L$ ratio in the IME model (Eq. 2), which reduces the impact of missing pixels in the masked plume, and by the $U_{\mathrm{eff}}$ term (Eq. 3), which is generated using realistic estimates of the retrieval noise.

### 3.2 Quantification of emission rates

The first test for the evaluation of the IME-based $Q$ quantification method has consisted in the comparison with the divergence integral (DI) method described in Chulakadabba et al. (2023). We have generated $Q$ estimates for a subset of 12 plumes from the RF06 campaign with the two methods. The same $\Delta$XCH$_4$ maps from the matched-filter retrieval were used as an input for the two methods, but each method was constrained with the wind data with which it is typically run (GEOS-FP for the IME-based method, and HRRR for the DI method).

The results from the quantification of the 12 plumes by the two methods are displayed in Fig. 6. Despite the different fundamental basis and wind data used by the two methods, we find a relatively good agreement in the quantification of those selected plumes with differences in $Q$ being typically below 20% for most of the plumes. Since the DI $Q$ estimation method has been thoroughly validated through independent controlled release tests (El Abbadi et al., 2024), this good agreement between





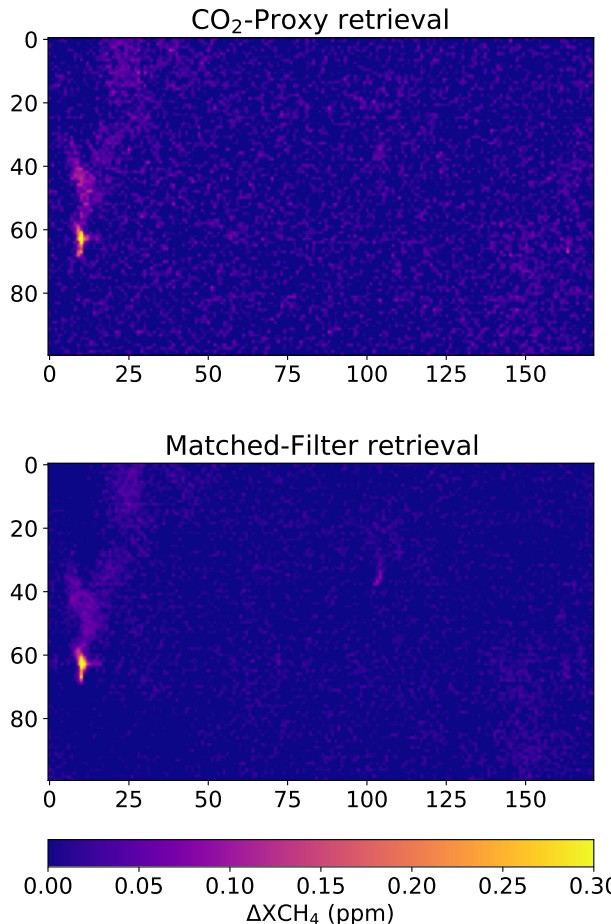

**Figure 4.** Comparison of $\Delta XCH_4$ maps generated with MethaneAIR's official CO2-proxy retrieval and the matched-filter retrieval proposed in this study. For the CO2-proxy $XCH_4$ retrieval, the $\Delta XCH_4$ map is generated as the per-pixel methane column mixing ratio ($XCH_4$) minus its mean value.

the two methods suggests that our implementation of the IME model for MethaneAIR, constrained with GEOS-FP winds, can reproduce the emission rates for the conditions of the RF06 Permian Basin campaign.

In order to further validate the plume detection and quantification skills of our processing chain, we have processed several
220    MethaneAIR acquisitions over a controlled methane release experiment on 25 October 2022 in Arizona (USA) (RF01E campaign, see Sec. 2.5). Results from the $\Delta XCH_4$ maps for three of the weakest releases detected during this experiment (metered values of 205, 96, and 63 kg/h) are shown in Fig. 7. Each map covers an area of about 2.5 km side. The maps show that the methane enhancements stand out from the background in all three cases, without systematic retrieval artefacts being present in the vicinity of the plume. Approximately the same number of pixels are affected by $\Delta XCH_4$ values above the noise level
225    for the $Q = 63$ kg/h and $Q = 96$ kg/h plumes. This could be due to the stronger wind during the weaker emission (0.9 versus





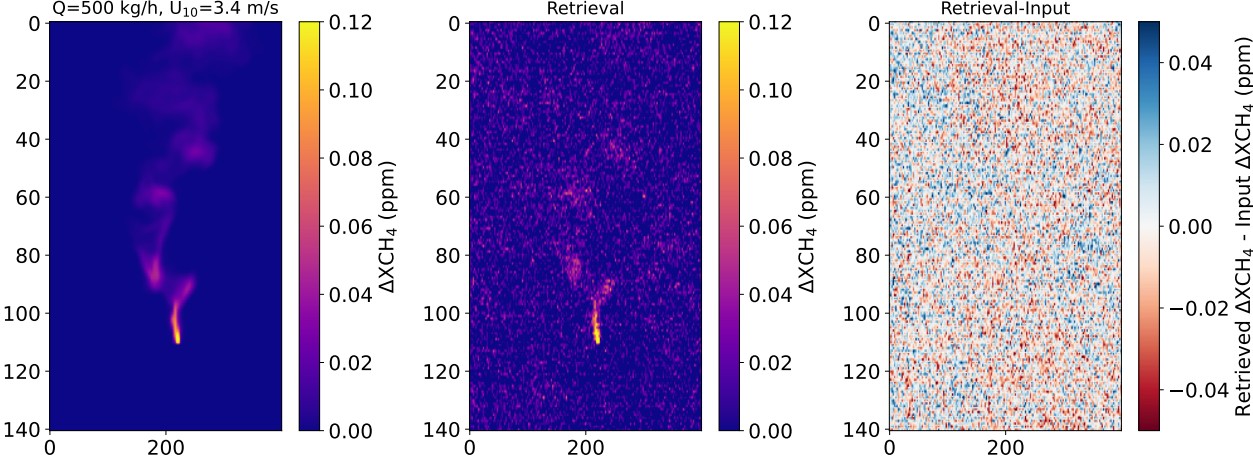

**Figure 5.** Results from end-to-end $\Delta XCH_4$ retrieval simulations for a $Q = 500$ kg/h plume embedded in a Permian Basin data granule from the RF06 campaign. The input WRF-LES plume is displayed on the left panel, the retrieved $\Delta XCH_4$ map on the central panel, and the difference between the two on the right.

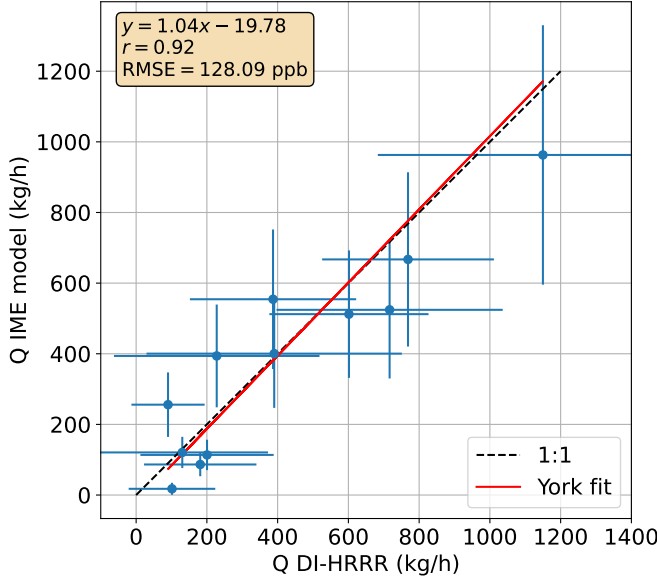

**Figure 6.** Comparison between $Q$ estimates obtained with the IME-based model used in this work (see Section 2.3) and the divergence integral method (DI) described in Chulakadabba et al. (2023) for 12 selected plumes from the RF06 campaign. Error bars represent the 1-sigma error for the IME $Q$ estimates, and the 95% confidence interval for the DI estimates.




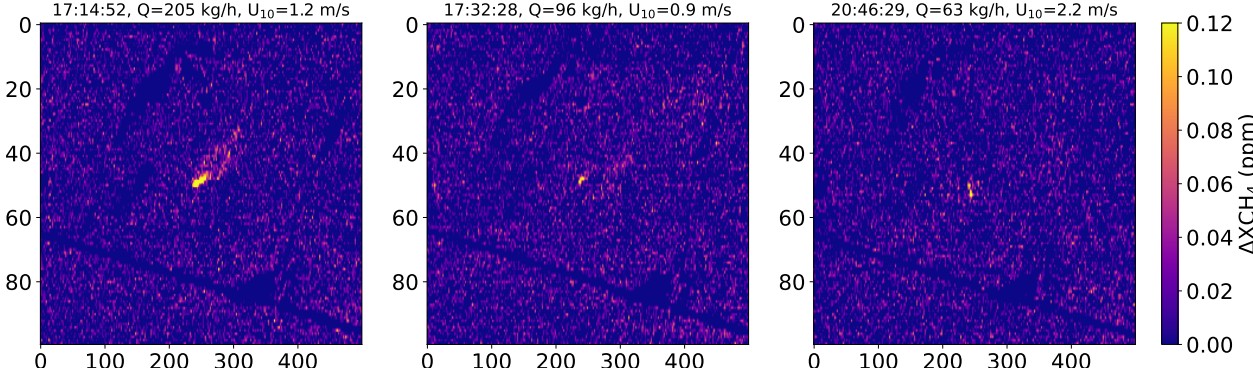

**Figure 7.** $\Delta$XCH$_4$ maps over the controlled methane release experiment in Arizona on 25 October 2022. Overpasses corresponding to relatively weak emissions have been chosen. The flux rate (Q) and 10-m wind speed (U$_{10}$) in the title of each panel correspond to the metered values.

2.2 m/s according to in situ measurements) originating a larger plume close to the source, which implies that the probability of plume detection is not always inversely proportional to wind speed, but in some cases there is an optimal wind speed for plume detection.

These results suggest that plume detection limits of about 60 kg/h could be achievable with MethaneAIR flying at 12 km above ground. However, two points must be noted. First, the location of the controlled release site is known beforehand, so the identification of the enhancement and its confirmation as a real plume is in this case much simpler than in the real case, where the location is unknown. Second, the plume detection process depends on several factors, including retrieval noise, occurrence of systematic errors, and wind speed. This causes that the "minimum detection limit", defined as the smallest source that can be detected in a given dataset, may substantially overestimate the plume detection capability of a sensor. The "probability of detection" concept, leading to continuous probability of detection functions expressing with which probability a plume of a given flux rate will be detected, can better represent the variability in detection limits found under normal operation conditions (e.g. Conrad et al., 2023). We will continue this discussion in Section 3.4.

The metered $Q$s from the controlled releases have been used for a first assessment of the performance of our IME-based $Q$ estimation model. The comparison between the metered values and the $Q$ estimates from our processing (matched-filter $\Delta$XCH$_4$ retrievals and IME-based $Q$ estimates constrained by GEOS-FP winds) are shown in Fig. 8. The results from MethaneAIR reproduce well the metered values ($r^2 =$0.96), for both high and low flux rate values (100–1000 kg/h range), which gives confidence in the performance of our entire processing chain. We acknowledge, however, that this sample only contains 8 points, and that a denser sampling over this site and others with different surface and wind conditions would be needed to extract more solid conclusions about the performance of our processing chain, similar to the more comprehensive analysis presented in Chulakadabba et al. (2023) and El Abbadi et al. (2024).



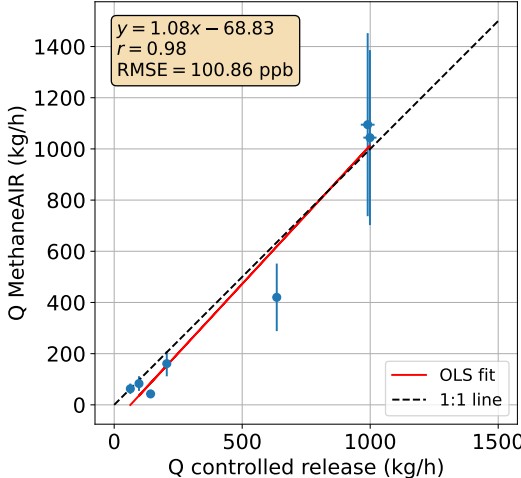

**Figure 8.** Comparison between metered flux rates from the controlled release experiment in Arizona on 25 October 2022, and flux rate estimates from MethaneAIR using the processing chain described in this work. Error bars in the $y$-axis represent the 1-sigma error for the IME $Q$ estimates from MethaneAIR.

## 3.3 Attribution of plumes to sources

MethaneAIR's nominal operation mode provides a native pixel size of $5.76 \times 25\,\mathrm{m}^2$, which is larger than the 1–5 m spatial sampling range often found for airborne spectrometers (El Abbadi et al., 2024). This coarser spatial sampling is selected for MethaneAIR in order to increase the area coverage of each overpass, which is required to evaluate area fluxes as well as point sources. However, MethaneAIR's spatial sampling is still usually sufficient to attribute the detected plumes to their sources. This is illustrated in Fig. 9, which shows examples of methane plumes represented on top of at-sensor radiance maps from the same MethaneAIR acquisitions from which the $\Delta$XCH$_4$ maps are derived. The analysis of the combined $\Delta$XCH$_4$ and radiance maps is often sufficient to identify the facilities responsible for each emission. However, the combination with infrastructure databases, such as the Oil and Gas Infrastructure Mapping database (OGIM) (Omara et al., 2023), and very high resolution optical imagery is needed to refine the information on the sources. Combining MethaneAIR radiance and $\Delta$XCH$_4$ maps with those external data sources, we attribute the plumes in Fig. 9 to different infrastructure elements. For example, plume # 1 comes from a complex wellpad; plume #2 from a compressor station; plume #3 from a pipeline, and plumes #4 to #7 from processing plants.

A zoom-in of Fig. 9's methane plume #7 is provided in Fig. 10. The plume is represented on top of a very high resolution satellite image downloaded from Google Maps. It is difficult to determine the exact source responsible of the emissions, but we discard the flare and the compressor units as potential sources as they are located elsewhere on the plant.



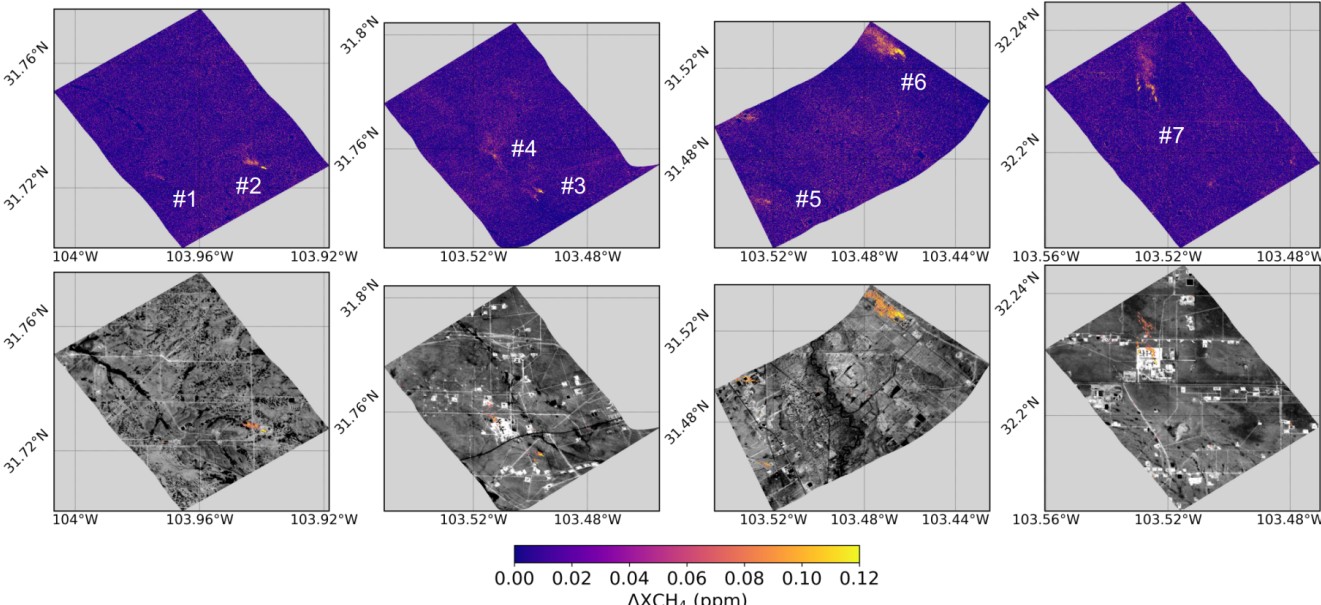

**Figure 9.** Sample methane plumes detected in $\Delta$XCH$_4$ maps derived from data subsets from the MethaneAIR RF06 Permian Basin campaign. The raw $\Delta$XCH$_4$ maps are shown in the top row, and the plumes represented on top of the radiance maps are presented in the bottom row. The numbers in white are used to refer to the different plumes in the text.

## 3.4 Large-scale $\Delta$XCH$_4$ mapping

We have assessed MethaneAIR's potential to survey methane point sources across large regions using entire flightlines from the RF06 and MX025 Permian Basin campaigns (see Section 2.5). The area covered by each flight (hundreds of kilometers in each case) is displayed in Fig. 11 with mosaics of near-infrared reflectance (at-sensor radiance at 1623 nm normalized by the top-of-atmosphere solar irradiance at the same wavelength). The detected methane sources and their intensity are depicted by red circles of varying size. It can be seen that the distribution of active sources varies considerably from one campaign to the other, as shown by the area marked with the blue rectangle. We note that the RF06 and MX025 sampling covers some of the most active oil and gas production regions in the Permian, contributing more than one-third of the total Permian oil and gas production in 2023 (Enverus Prism, 2024). In addition, between 2021 and 2023, oil and gas production increased by 32% and 40% in RF06 and MX025, respectively. Furthermore, both RF06 and MX025 are active gas flaring regions in the Permian. We suggest that such increased oil and gas activity could lead to increased emissions plausibly due to increased stress on the gathering and processing segments, especially if their processing capacity did not increase accordingly.

A more quantitative view on the detected points sources is provided in Fig. 12, which represents the distribution of emission rates obtained from all the plumes that have been detected and quantified in the RF06 and MX025 datasets. This figure shows the higher number of plumes detected in the RF06 dataset with respect to MX025 (121 and 78, respectively). We also find



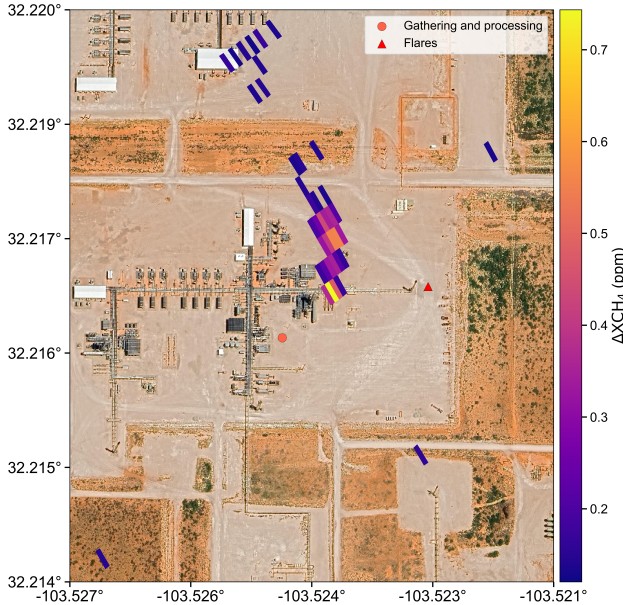

**Figure 10.** Methane plume from MethaneAIR represented on top of a high resolution image showing the facility responsible for the emissions. The methane plume corresponds to plume #7 in Fig. 9. The background image was downloaded from © Google Maps, and was acquired by Airbus in 2023.

a difference in the minimum flux rates found in each dataset, with the smallest flux rates in the range of 25 kg/h for RF06, and 100 kg/h for MX025 (see inset of Fig. 12). In addition to inter-annual variations in oil and gas production, external factors affecting our ability to detect and quantify methane plumes with MethaneAIR may partly explain this difference. In particular, wind speed is an important driver for plume detection (Ayasse et al., 2023). The GEOS-FP wind product shows average wind speeds of about 3.5 m/s for RF06, whereas stronger winds of about 5 m/s are reported in GEOS-FP during the MX025 flights, which may have led to higher detection limits for this campaign. Finally, Fig. 12 shows that 3 plumes above 1500 kg/h could be detected in MX025, although the number of plumes above 1000 kg/h is similar for the two datasets (5 for RF06 and 6 for MX025). A potential offset in plume quantification caused by differences in acquisition conditions for the two campaigns can also not be ruled out to explain the observed differences in the flux rate distributions. The same trends (greater number of detections for RF06, with higher flux rate peak values and detection limits for MX025) are also found for the official collection of RF06 and MX025 plumes generated by MethaneAIR's data processing platform and made available to users (MethaneSAT Science Team, 2024), although the number of plumes from RF06 and MX025 in that collection is about 25% that of the generated in this work.

We have further analysed the plume detection limits of MethaneAIR for the Permian Basin using the data from the RF06 and MX025 campaigns. As mentioned earlier in this work, the detection of a plume in a $\Delta XCH_4$ map depends on several factors,




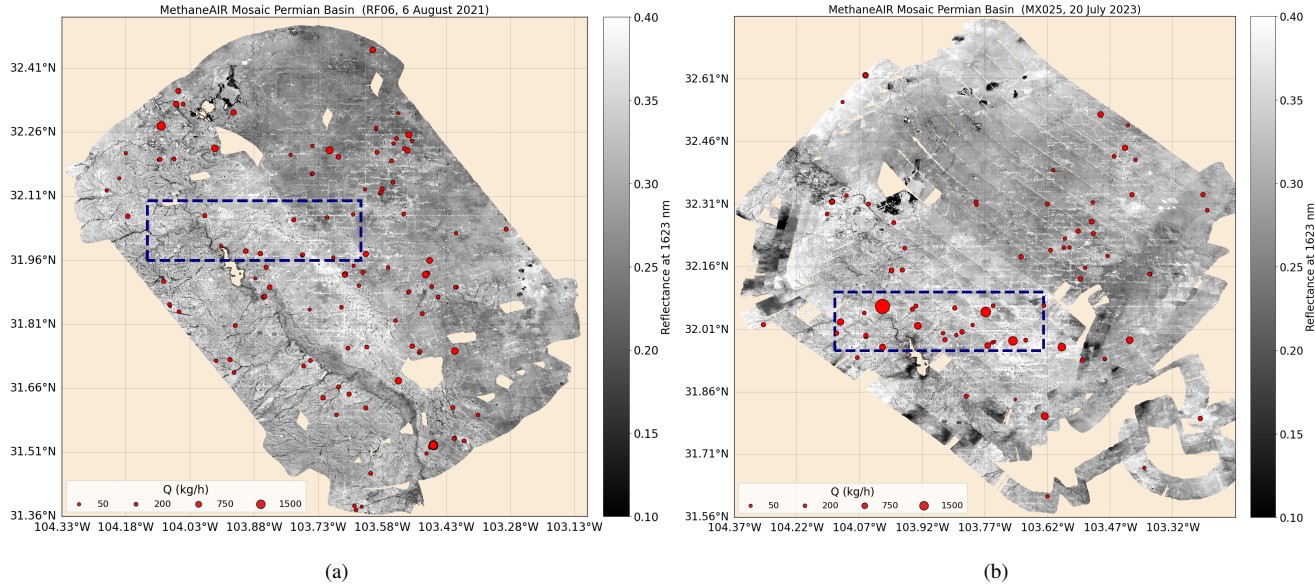

(a)                                                                                      (b)

**Figure 11.** Composite of at-sensor reflectance data showing the areas in the Permian Basin covered by the MethaneAIR campaigns RF06 (a) and MX025 (b). At-sensor reflectance is calculated as the at-sensor radiance at 1623 nm normalized by the top-of-atmosphere solar irradiance at the same wavelength. The red circles depict the methane plumes detected for each campaign. The blue rectangle depicts an area with strong changes in emission activity between the two dates.

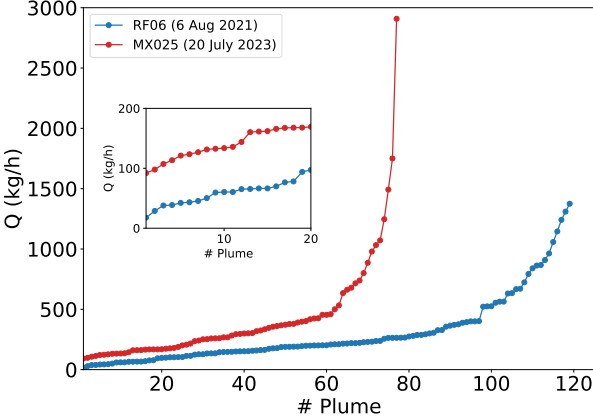

**Figure 12.** Summary of the flux rates ($Q$) estimated from the methane plumes detected in the RF06 and MX025 datasets (red circles in Fig. 11). The inset shows a zoom-in of the plumes with the smallest $Q$s. Uncertainties in the single $Q$ estimates are not represented for visibility purposes.



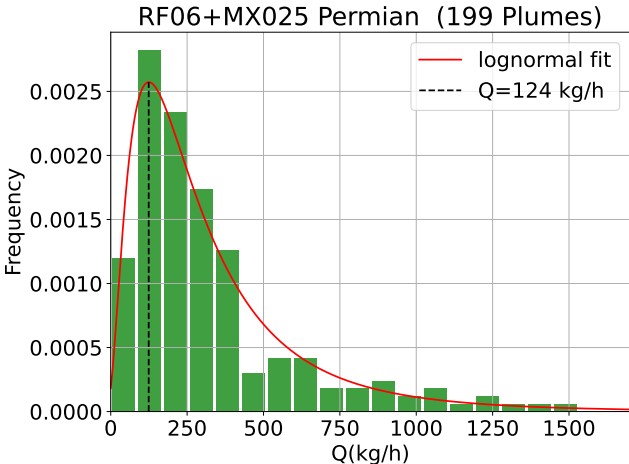

**Figure 13.** Histogram of the flux rates ($Q$) estimated from the plumes detected in the RF06 and MX025 datasets. The two campaigns have been combined in order to increase the plume sample. The dash line marks the $Q$ value for which the distribution of estimated $Q$s deviate from a power law, which can be interpreted as a rough estimation of the source detection limit processing MethaneAIR data from the Permian Basin using the processing scheme proposed in this work.

including the wind speed, the retrieval noise introduced by the surface albedo, or an enhanced spatial variability of $\Delta XCH_4$ caused by neighbouring sources. Therefore, a parametric probability distribution function (PDF) depending on those factors would be needed to determine the probability of detection (POD) of any given plume. For example, Conrad et al. (2023) built

295 such PDF (depending on several parameters, including wind speed) for several airborne sensors using about 500 controlled releases, leading to distributions of true positive and false negative detections that could be used as a reference distribution to fit a parametric model. Ayasse et al. (2023) used a similar approach to assess the POD of the AVIRIS-NG/CAO systems. In the case of Bruno et al. (2024), they assessed GHGSat-C1's POD fitting a sigmoid function to a range of WRF-LES plumes recreating different plume intensities and morphologies.

300     In our case, however, we do not have a reference emission distribution dataset that we can use to fit a POD model for our MethaneAIR processing chain. As an alternative, we obtain an estimate of MethaneAIR detection limits for the Permian Basin by simply examining the shape of the emission distribution curve that we obtain from combining the RF06 and MX025 plume datasets. We adopt as detection limit the flux rate at which the emission distribution curve (modelled as a lognormal function) starts to deviate from the monotonous increase trend (typically in the form of a power law) which would be expected if all

305 plumes were detected. The result from this analysis is shown in Fig. 13. We find that the flux rate at which the distribution of MethaneAIR plumes deviates from the power-law trend is about 124 kg/h. We can expect that the majority of sources above this threshold would be detected in the RF06 and MX025 datasets. Actually, this number may change if the RF06 and MX025 datasets were analysed separately (with a lower number for RF06, and a higher number for MX025). However, the independent analysis of the two datasets is difficult because the single datasets are too small for a robust lognormal fit.





## 4 Conclusions

We have developed a processing chain for the detection and quantification of point source methane emissions with the MethaneAIR airborne spectrometer. Our goal was to implement a computationally-efficient retrieval able to maximize the probability of plume detection. We have achieved those goals by combining a data-driven $\Delta XCH_4$ retrieval, based on the matched-filter concept, with a plume detection and segmentation approach based on visual inspection of the resulting $\Delta XCH_4$ maps. Flux rates are estimated from the detected plumes using an IME-based method. This processing scheme enabled the analysis of methane point sources across the Permian Basin using data from two campaigns in 2021 and 2023.

We have shown the potential of the matched-filter retrieval for high spectral resolution measurements in the 1650 nm window. The results from our matched-filter $\Delta XCH_4$ retrieval compare well with those from the physically-based $CO_2$-proxy $XCH_4$ retrieval used in MethaneAIR's operational processing chain. The matched-filter retrieval can only provide $XCH_4$ enhancements, and is therefore not an alternative to the $CO_2$-proxy $XCH_4$ retrieval, which does provide the total $XCH_4$ column content required to evaluate area emissions. However, the $\Delta XCH_4$ retrieval by the matched-filter is of simple implementation and computationally efficient, and offers a lower retrieval noise than the $CO_2$-proxy $XCH_4$ retrieval, which is advantageous for point source work.

Our results from the processing and analysis of two MethaneAIR flights over the Permian Basin show the potential of MethaneAIR for the detection and quantification of methane point sources across large areas, with about 120 plumes being detected in the 2021 flight, about 80 in the 2023 flight, resulting in a combined detection limit for which most of the plumes would be detected of about 124 kg/h. We attribute part of the differences in the number of plumes detected from each flight to changes in oil and gas production in the region over time. However, we also acknowledge that the differences in the number of plume detections can also be due to different data acquisition conditions. In particular, the stronger winds found in 2023 with respect to 2021 may have led to the greater detection limits in 2023, which is also consistent with the findings by other authors (Ayasse et al., 2023).

We have opted for a manual plume detection and segmentation approach in order to ensure that the maximum number of plumes could be detected. However, this step introduces the need for a human-in-the-loop in our processing chain, which challenges its application to large volumes of data despite the improvement in processing time enabled by the matched-filter. Machine-learning based plume detection approaches (e.g. Růžička et al., 2023) could help reduce the need for human supervision, although the implementation of a fully-automated processing chain is challenging if both the detection limits and the probability of false positives are to be kept to a minimum, as it was the goal in this work.

Overall, the computationally-efficient approach described here as applied to MethaneAIR measurements can also be extended to MethaneSAT in order to help advance the point source detection capacity, as the spectral characteristics are very similar between the airborne and satellite platforms.

 

*Author contributions.* LG led the study, developed the data processiing chain, and wrote the paper, incorporating comments and revisions from all authors. AC and MS contributed DI and IME-based flux rate estimates. JR contributed to the implementation of the matched-filter retrieval. JW, MO, RG supported the identification of active sources and the interpretation of emissions from the Permain Basin. MS, JEF AND SCW were in charge of campaign planning, instrument calibration and data processing.

345 *Competing interests.* We declare no competing interests.

*Acknowledgements.* The Environmental Defense Fund provides primary support for the MethaneAIR and MethaneSAT projects to Harvard University.



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
