# Peer review of "Remote sensing of methane point sources with the MethaneAIR airborne spectrometer"

_EGUsphere, 2024_

## Author Comment (AC1)

**Reviewer #2**

The authors provide CH4 retrieval and emission quantification methods for the MethaneAIR imaging spectrometer based on a matched filter and integrated mass enhancement method. The manuscript is well written and provides interesting results. The manuscript does not have a code and data availability section. I have some comments on the methodology that should be addressed for me to recommend the paper for publication.

*Thank you for the positive comments and the very careful review.*

**Introduction**

**L20ff**: The grouping of methane imagers based on 1600 nm and 2300 nm windows is a bit arbitrary. I would argue that the main difference between AVIRIS and MAMAP-2D instruments are the difference in spatial and spectral resolution.

*We agree, and that is indeed the rationale for the split between 1600 and 2300 nm instruments in the text, as mentioned in e.g. "First, we have the spectrometers sampling the entire solar spectrum ($\sim$400--2500\,nm) with a relatively coarse spectral sampling between 5 and 10\,nm, and a relatively high spatial resolution (a few meters in the case of some airborne instruments)".*

*We have now emphasized the typically coarser spatial sampling of instruments relying on the 1650 nm band for methane retrievals: "The second group of methane-sensitive spectrometers sample a narrow spectral window around the 1650\,nm methane absorption, with a sub-nanometer spectral sampling, and a typically coarser spatial sampling".*

**L40f:** Please define area sources. Is a landfill already an area source?

*Definition added to the first sentence of the Introduction, as "...methane emissions from small infrastructure elements, also known as point sources"*

**L47ff:** Maybe already explain here why CO2-proxy retrievals are less precise than matched filters.

*Clarification added as "Also, the normalization of the retrieved methane column density by the per-pixel XCO$_2$ proxy increases the 1-$\sigma$ error of the resulting XCH$_4$ maps, which may lead to higher plume detection limits."*

**Method**

**Figure 1:** Instead of arbitrary spectra for CH4, CO2 and H2O, it would be nice show spectra for typical atmospheric concentrations.

*Figure and caption have been updated to include the column concentrations for each gas.*

**L86ff:** Foote et al. (2020) introduces an albedo correction term to remove systematic errors in XCH4 plumes due to deviations between the mean spectrum and the local spectrum. The systematic errors are likely to introduce systematic errors in the emission estimates. I think it is necessary to test if the albedo correction affects the results.

*We would argue that MethaneAIR's high spectral resolution enables a better decoupling of methane and surface reflectance/albedo than what is possible with coarser spectral resolution instruments, such as the AVIRIS-NG spectrometer used by Foote et al.. This would make the albedo correction less relevant. Also, the topic of the impact of surface albedo on the matched-filter retrieval is already tackled by the discussion around Fig. 3. For this reason, we prefer not to make a relatively*

major extension to the study by implementing and evaluating Foote's albedo correction in our retrieval.

This clarification has been added: *"We expect that MethaneAIR's high spectral resolution enables a better decoupling of methane and surface reflectance in the retrieval than what is usually found in coaser spectral resolution retrievals \citep{AYASSE2018386}."*

**L95**: How do you account for varying observation angles and surface elevation during data acquisition?

This has been clarified as *"In the case of the target spectrum $\vec{k}$, this is calculated at high spectral resolution from pre-computed transmittance spectra stored in a look-up table (LUT). For that, we interpolate the LUT considering the mean value of the sun zenith angle and the ground-to-sensor distance within each data granule, whereas a per-column view zenith angle is used in order to account for across-track gradients in the observation angle. It must be stated that local gradients in surface elevation are not accounted for by this approach."*

**L101f:** The small number of samples also affect the mean vector. Did you test the effect of computing the mean vector for a larger sample on your retrieval?

No, we didn't, but we expect the largest effect to be on the covariance matrix.

**L114f**: Kuhlmann et al. (2024, https://doi.org/10.5194/egusphere-2024-3494) identified CH4 emissions from vent stack in Romania using AVIRIS-NG that were not visible in high-resolution images. How many plumes did you reject, because they are not linked to any infrastructure, and do you see the possibility that you miss such sources in your analysis?

Thanks to MethaneAIR's high spectral resolution, the large majority of the plumes we derived from MethaneAIR where clear enough to have confidence in the detection, making the need for cross-checking with very high resolution imagery to be very small.

This paragraph reads now: *"the candidate plumes identified through a first screening based on visual inspection are compared with the input spectral radiance data at the continuum of the 1650\,nm absorption feature to discard false positives due to surface patterns(e.g. clouds). However, thanks to MethaneAIR's high spectral resolution, the large majority of the plumes we derived from MethaneAIR were clear enough to have confidence in the detection, making the need for cross-checking with very high resolution imagery very small."*

**L113:** Do you use the plume length or the square root of the detectable plume area as length scale

Information added as *"where the plume length $L$ is approximated by the square root of the detectable plume."*

**L128ff:** Effective wind speed also depends on emission height and vertical mixing. Maasakkers et al. (2022) derive their empirical equation for a landfill, which I would assume, emits near the surface, while emissions from oil and gas can be elevated from vent stacks or on top of processing facilities. How do you account for this in your method?

Thanks for this interesting point. The height of the source may indeed have an impact on the IME model, but in general there is no information on source height that we could use to contrain an IME model with this dependency during the operational processing.

We have specified in the text that the IME model from Maasakkers et al. was derived for surface-level emissions ("*which was proposed by \citet{bram_landfills} for GHGSat for surface-level emissions (landfills in their case)*").

**L139f:** Please provide more information about the DI method.

The following lines have been included in a new section "*2.4 Reference plume quantification methods*":

"*For the DI method, we calculate the fluxes along rectangular boxes around the source of interest. First, we compute the flux for each pixel along the chosen rectangular box. We then determine the gradient of XCH$_4$ and multiply it by the wind vector at each pixel. Based on Green's theorem, we sum all the fluxes to obtain the total flux for a given rectangle. By repeating this calculation for rectangles of different sizes around the source, we obtain a statistical estimate of the flux around the source of interest. In other words, we sample the flux spatially across the observing region using the DI method. Unlike the IME method, we neither sum all the pixels within the plume nor use an effective wind speed.*"

**L143ff:** Section 2.4 does not provide enough information to judge the accuracy of the end-to-end simulator. I would assume that it does not include systematic errors in the plume, which might explain why Figure 5 shows good agreement between retrieval and input. I suggest to either remove the end-to-end simulator from the manuscript or provide more details including a more detailed analysis, which should be quite interesting.

The following paragraph has been added to provide more information about the simulation approach: "*The spatially-distributed \dx\ values from the simulated plumes were converted into per-pixel plume transmittance spectra with the same LUTs used for the generation of the $\vec{k}$ spectrum, which is an input to the \dx\ retrieval. With this approach of using the same radiative transfer scheme for the forward simulations and for the \dx\ retrieval, we avoid introducing uncontrolled systematic errors in the end-to-end simulation framework (e.g. as from different gas vertical profiles).*"

**Results**

**Figure 2:** Please add a (rough) scale to the image.

A scale bar has been added,

**L182ff / Figure 4:** I really would like to see the difference between proxy and matched filter (as in Fig. 5). Do you find systematic differences between the methods, in particular inside the plume, what might be the reason, and how would they affect your emission estimate?

Fig. 4 has been updated to show the difference map, and these lines have been added to the main text "*Two small clusters of pixels with systematic offsets can be seen in the difference map, at pixel coordinates (10, 60) and (10, 40) corresponding to the larger plume in the subset. However, these enhancements are close to the noise level and have a different sign, leading to an almost zero offset when aggregated to calculate the IME and, subsequently, $Q$.*"

**L198ff:** (see my previous comment on the end-to-end simulator)

Please, see reply to **L143ff.**

**L207ff:** Please provide more information how the DI method has been implemented in this study. Following Chulakadabba et al. (2023), the DI method sums over all pixels along rectangular for difference from the source location to the edge of the detectable plume (except for subtracting the background, which would be about zero for the matched filter). This isn't much different from the IME method, which sums over all pixels in the detectable plume, while excluding the background (which is close to zero).

The main difference between IME and DI method seems to be the effective wind speed. What wind speed is used for the DI method and how does compare to effective wind speed used with the IME method? Would the differences vanish if the same data source (GEOS-FP or HRRR) for the wind speed is used for both methods?

Please, see reply to **L139f** regarding the description of the DI method.

We fully agree with the reviewer in that variations in wind speed data will proportionally affect Q estimates because of the linear relationship between U10 and Q in the IME model (Eq. 2). In this study, our intention was not to evaluate the IME or DI formulations per-se, but to show the validity of our basic IME+GEOS-FP implementation for the estimation of flux rates from the large dataset of plumes detected from the matched-filter output. The DI model constrained with HRRR winds offers an accurate Q estimation framework (although less practical for the processing of large datasets) that we took as a reference for the evaluation of our IME-based Q estimates.

This is discussed in the text as *"each method was constrained with different wind data: the IME-based method is run with GEOS-FP data, as this is the configuration that we apply for the processing of the large plume datasets derived in this work, whereas the DI method is constrained with HRRR wind data, as this is the configuration that potentially provides the most accurate reference for intercomparison with the IME approach"*

**L223f:** It would interesting do discuss the potential for systematic errors inside the plume here (see my previous comment).

Please, see reply to **L182ff / Figure 4.**

**L226f:** I do not see why the detection limit should not always increase with wind speed. However, detection limit can depend on other factors such as turbulence.

Clarification added as: *"which implies that the probability of plume detection is not always inversely proportional to wind speed, but in some cases there is an optimal wind speed for plume detection: low-to-moderate winds enabling the development of a plume covering several pixels with \dx\ values above the noise level."*

**L238f:** What is the uncertainty of the emission rates from the controlled releases?

The caption of Fig. 8 has been updated as *"The metered flux rates correspond to 30-second averages. Error bars in the $y$-axis represent the 1-sigma error for the IME $Q$ estimates from MethaneAIR, and error bars in the $x$-axis represent the standard deviation in the metered flux rate values in the 30-second window."*

**Figure 8:** Why is the RMSE measured in ppb?

Thanks for pointing this out, the legend has been corrected to kg/h (also in Fig. 6)

**L280ff:** Does the wind speed vary spatially and temporally in the campaign area during data acquisition? I expect that would affect the detection limit during the campaign.

*We agree. A clarification has been added as "The GEOS-FP wind product shows average wind speeds of about 3.5\,m/s for RF06, whereas stronger winds of about 5\,m/s are reported in GEOS-FP during the MX025 flights, with a standard deviation of 0.5\,m/s in both cases. The stronger winds may have led to higher detection limits for the MX025 campaign. We have not analyzed spatial and temporal variations of wind speed during data acquisition for each campaign in depth, but such changes would also have an impact on plume detections in each campaign"*

**Figure 12:** Do you see a dependency of flux rates on wind speed?

*We would need a more careful analysis, but from a simple representation of the data included in Fig. 12, we do not see a depency of the flux rate estimates on u10 (see figure below).*

[Figure]

**L292:** Does the albedo change between the campaigns?

*This line has been rephrased as "As mentioned earlier in this work, the detection of a plume in a \dx\ map depends on several factors, including the wind speed, the retrieval noise (driven by at-sensor radiance and local variability in the surface albedo), and the modification of \dx\ gradients by neighboring sources."*

**L292f:** It is unclear what do you mean with "enhanced spatial variability of ΔXCH4".

*The "enhanced spatial variability of ΔXCH4" has been rephrased as "the modification of \dx\ gradients by neighboring sources"*

**Conclusions**

**L312f:** Please specify why a computationally efficient retrieval would improve the detection limit.

*Sentence rephrased as "Our goal was to implement a \dx\ retrieval which was both computationally-efficient and able to maximize the probability of plume detection"*

**L332:** I suggest adding that a major advantage is minimizing the number of false positives.

*Statement added as ", with a minimum rate of false positives"*

---

## Author Comment (AC2)

**Response to reviewers (egusphere-2024-3577)**

Please, find enclosed our point-by-point responses to the comments and suggestions made by the two reviewers of our manuscript. The comments from the reviewers are in black, and our responses are in blue.

Based on the reviewers' comments, the main changes to the manuscript after revision are:

- Figures 1, 2, 6, and 8 have been corrected
- An extra panel has been added to Fig. 4 to show the difference map between the matched-filter and the CO2-proxy retrievals
- The descriptions of the end-to-end simulation approach and the DI emission rate estimation method have been improved
- A comparison with the MethaneAIR Level-4 point-source product has been added
- The lists of plume locations and flux rates generated during this study are now provided as supplement materials

**Reviewer #1**

The authors provide a straightforward reanalysis of MethaneAir data that has been processed with a more point source centric algorithm. The paper is clear, well written, and very interesting to see a matched filter algorithm applied to a high spectral resolution instrument like MethaneAir. I do have a few comments, most important Comment #4, as I think these results have broad implications around the importance of point sources generally. I ask the authors to clarify and provide this context before I recommend for submission.

We would like to thank you for the positive and thoughtful comments.

1. Line 145. Can you confirm or clarify how the injection of WRF-LES concentrations was performed? You calculate transmission due to extra CH4 column concentration and then apply to MethaneAir radiance?

Yes, that was indeed the approach.

The following paragraph has been added: "*The spatially-distributed \dx\ values from the simulated plumes were converted into per-pixel plume transmittance spectra with the same LUTs used for the generation of the $\vec{k}$ spectrum, which is an input to the \dx\ retrieval. With this approach of using the same radiative transfer scheme for the forward simulations and for the \dx\ retrieval, we avoid introducing uncontrolled systematic errors in the end-to-end simulation framework (e.g. as from different gas vertical profiles).*"

2. Line 194. Why aren't matched filter retrievals suitable for estimation of total area budgets? Because the background normalization in an MF algorithm "removes" regional gradients? Ultimately in an area flux inversion, one has to create XCH4 enhancements relative to the background for assimilation. If one plotted retrieved XCH4 enhancements derived from matched filters vs. CO2 proxy, and they correlated reasonably well, I don't see why a matched filter algorithm couldn't be used. Please explain.

Thanks for this good point. Indeed, the matched-filter retrieval could theoretically be used for area flux inversions as well, but we would need a number of tests to better assess this posibility. One potential limitation of our implementation of the matched-filter retrieval to generate XCH4 data for area inversions would be the neglection of topographic affects.

We have added this discussion to the text: "*Nevertheless, physically-based total-column XCH$_4$ retrievals from the CO$_2$-proxy (as opposed to the data-driven \dx\ retrievals by the matched-filter) are preferred for the estimation of area- and total-emission budgets, which is a key application of MethaneAIR. A physically-based pixel-wise XCH$_4$ retrieval can better account for spatial gradients in the methane background caused by atmospheric transport and topography. This implies that the matched-filter \dx\ output is currently not an alternative to the CO$_2$-proxy XCH$_4$ retrieval for the calculation of area and total methane fluxes from MethaneAIR data cubes.*"

3. Figure 8. Why are only 8 data points shown here, when El Abbadi et al. (2024) reports 24 controlled releases were performed for MethaneAir? Shouldn't all points be shown? Were there plumes that didn't perform well with this new algorithm applied, hence they are not shown?

We have clarified this point with the following text: "*we processed data from another research flight, RF01E, which was carried out on 25 October 2022 over a single-blind volume-controlled*

*methane-release experiment near Phoenix (USA) \citep{ju_mair_2023}, resulting in 8 match-ups between MethaneAIR acquisitions and controlled releases. This controlled-release campaign included a second day on 29 October 2022 \citep{ju_mair_2023}, but we chose to focus our analysis on the 25 October flight because of the more stable winds and the smaller plumes. These conditions enabled both a sufficient sample of reliable match-ups and the evaluation of our plume detection limits."*

4. Please include point source datasets as part of the SI

We will provide the point source datasets for the RF06 and MX025 flights that have been generated in this study as supplementary materials.

5. Related to comment #4 - I am curious about how the improvement on detection limit affects the general understanding of point vs area sources, which appears to be a central mission thrust of MethaneAir. Looking at other datasets that are available online (https://showcase.earthengine.app/view/methanesat), I count 28 point sources that were detected from RF06, while this study reports 121. Relatedly, that dataset on Google Earth Engine states that point sources make up 33,700 kg/h compared to a total flux of 91,000 kg/h (37%). How much methane total do you now quantify from point sources using this new matched filter algorithm? It appears that it would have to be higher, potentially much higher. As some bottom-up studies have leveraged MethaneAir to suggest a small contribution from emission sources above 100 kg/h (e.g., https://doi.org/10.5194/egusphere-2024-1402), it appears that the conclusions from those studies may have been an artifact of point source detection limit. Though it is out of scope for this paper to comment on those studies, it is appropriate for you to state how much total CH4 there is from the Permian scenes you processed, and how that relates to total fluxes derived from the CO2 proxy method.

Thanks for this comment. We agree that better understanding the contribution of individual point sources in regional-level emissions is important to define and guide potential mitigation approaches, although we consider that the discussion of regional fluxes may indeed be out of the scope of this work.

Following the suggestion by the reviewer, we have included a high-level comparison between the total emission rates from the plumes detected in this study with those from the MethaneAIR L4-DI (point sources) product which is available to users via Google Earth Engine (please, see text below).

Please, note, that the total rates that are obtained from the corresponding RF06 L4-DI file (https://developers.google.com/earth-engine/datasets/catalog/EDF_MethaneSAT_MethaneAIR_L4point#table-schema) is 26.7 t/h, and not the 33.7 t/h number which is indeed shown in the GEE website (https://showcase.earthengine.app/view/methanesat). We are in the process of interpreting and potentially correcting the second number.

*"These patterns are consistent with those found in the official MethaneAIR Level-4 product made available to users \citep{l4_di_plumes_2024}, namely a greater number of detections in RF06, and higher flux rate peak values and detection limits in MX025 (29 plumes and a minumum flux rate of 228\,kg/h for RF06, and 19 and 492\,kg/h for MX025). The total emissions calculated from the Level-4 dataset are 26.7\,t/h for RF06 and 25.6\,t/h for MX025, which is consistent with the 29 (25--34) t/h and 29 (23--36) t/h that we obtain from our dataset after filtering for plumes with flux rates above 200\,kg/h"*

**Reviewer #2**

The authors provide CH4 retrieval and emission quantification methods for the MethaneAIR imaging spectrometer based on a matched filter and integrated mass enhancement method. The manuscript is well written and provides interesting results. The manuscript does not have a code and data availability section. I have some comments on the methodology that should be addressed for me to recommend the paper for publication.

*Thank you for the positive comments and the very careful review.*

**Introduction**

**L20ff**: The grouping of methane imagers based on 1600 nm and 2300 nm windows is a bit arbitrary. I would argue that the main difference between AVIRIS and MAMAP-2D instruments are the difference in spatial and spectral resolution.

*We agree, and that is indeed the rationale for the split between 1600 and 2300 nm instruments in the text, as mentioned in e.g. "First, we have the spectrometers sampling the entire solar spectrum ($\sim$400--2500\,nm) with a relatively coarse spectral sampling between 5 and 10\,nm, and a relatively high spatial resolution (a few meters in the case of some airborne instruments)".*

*We have now emphasized the typically coarser spatial sampling of instruments relying on the 1650 nm band for methane retrievals: "The second group of methane-sensitive spectrometers sample a narrow spectral window around the 1650\,nm methane absorption, with a sub-nanometer spectral sampling, and a typically coarser spatial sampling".*

**L40f:** Please define area sources. Is a landfill already an area source?

*Definition added to the first sentence of the Introduction, as "...methane emissions from small infrastructure elements, also known as point sources"*

**L47ff:** Maybe already explain here why CO2-proxy retrievals are less precise than matched filters.

*Clarification added as "Also, the normalization of the retrieved methane column density by the per-pixel XCO$_2$ proxy increases the 1-$\sigma$ error of the resulting XCH$_4$ maps, which may lead to higher plume detection limits."*

**Method**

**Figure 1:** Instead of arbitrary spectra for CH4, CO2 and H2O, it would be nice show spectra for typical atmospheric concentrations.

*Figure and caption have been updated to include the column concentrations for each gas.*

**L86ff:** Foote et al. (2020) introduces an albedo correction term to remove systematic errors in XCH4 plumes due to deviations between the mean spectrum and the local spectrum. The systematic errors are likely to introduce systematic errors in the emission estimates. I think it is necessary to test if the albedo correction affects the results.

*We would argue that MethaneAIR's high spectral resolution enables a better decoupling of methane and surface reflectance/albedo than what is possible with coarser spectral resolution instruments, such as the AVIRIS-NG spectrometer used by Foote et al.. This would make the albedo correction less relevant. Also, the topic of the impact of surface albedo on the matched-filter retrieval is already tackled by the discussion around Fig. 3. For this reason, we prefer not to make a relatively*

major extension to the study by implementing and evaluating Foote's albedo correction in our retrieval.

This clarification has been added: "*We expect that MethaneAIR's high spectral resolution enables a better decoupling of methane and surface reflectance in the retrieval than what is usually found in coaser spectral resolution retrievals \citep{AYASSE2018386}.*"

**L95**: How do you account for varying observation angles and surface elevation during data acquisition?

This has been clarified as "*In the case of the target spectrum $\vec{k}$, this is calculated at high spectral resolution from pre-computed transmittance spectra stored in a look-up table (LUT). For that, we interpolate the LUT considering the mean value of the sun zenith angle and the ground-to-sensor distance within each data granule, whereas a per-column view zenith angle is used in order to account for across-track gradients in the observation angle. It must be stated that local gradients in surface elevation are not accounted for by this approach.*"

**L101f:** The small number of samples also affect the mean vector. Did you test the effect of computing the mean vector for a larger sample on your retrieval?

No, we didn't, but we expect the largest effect to be on the covariance matrix.

**L114f**: Kuhlmann et al. (2024, https://doi.org/10.5194/egusphere-2024-3494) identified CH4 emissions from vent stack in Romania using AVIRIS-NG that were not visible in high-resolution images. How many plumes did you reject, because they are not linked to any infrastructure, and do you see the possibility that you miss such sources in your analysis?

Thanks to MethaneAIR's high spectral resolution, the large majority of the plumes we derived from MethaneAIR where clear enough to have confidence in the detection, making the need for cross-checking with very high resolution imagery to be very small.

This paragraph reads now: "*the candidate plumes identified through a first screening based on visual inspection are compared with the input spectral radiance data at the continuum of the 1650\,nm absorption feature to discard false positives due to surface patterns(e.g. clouds). However, thanks to MethaneAIR's high spectral resolution, the large majority of the plumes we derived from MethaneAIR were clear enough to have confidence in the detection, making the need for cross-checking with very high resolution imagery very small.*"

**L113:** Do you use the plume length or the square root of the detectable plume area as length scale

Information added as "*where the plume length $L$ is approximated by the square root of the detectable plume.*"

**L128ff:** Effective wind speed also depends on emission height and vertical mixing. Maasakkers et al. (2022) derive their empirical equation for a landfill, which I would assume, emits near the surface, while emissions from oil and gas can be elevated from vent stacks or on top of processing facilities. How do you account for this in your method?

Thanks for this interesting point. The height of the source may indeed have an impact on the IME model, but in general there is no information on source height that we could use to contrain an IME model with this dependency during the operational processing.

We have specified in the text that the IME model from Maasakkers et al. was derived for surface-level emissions ("*which was proposed by \citet{bram_landfills} for GHGSat for surface-level emissions (landfills in their case)*").

**L139f:** Please provide more information about the DI method.

The following lines have been included in a new section "*2.4 Reference plume quantification methods*":

"*For the DI method, we calculate the fluxes along rectangular boxes around the source of interest. First, we compute the flux for each pixel along the chosen rectangular box. We then determine the gradient of XCH$_4$ and multiply it by the wind vector at each pixel. Based on Green's theorem, we sum all the fluxes to obtain the total flux for a given rectangle. By repeating this calculation for rectangles of different sizes around the source, we obtain a statistical estimate of the flux around the source of interest. In other words, we sample the flux spatially across the observing region using the DI method. Unlike the IME method, we neither sum all the pixels within the plume nor use an effective wind speed.*"

**L143ff:** Section 2.4 does not provide enough information to judge the accuracy of the end-to-end simulator. I would assume that it does not include systematic errors in the plume, which might explain why Figure 5 shows good agreement between retrieval and input. I suggest to either remove the end-to-end simulator from the manuscript or provide more details including a more detailed analysis, which should be quite interesting.

The following paragraph has been added to provide more information about the simulation approach: "*The spatially-distributed \dx\ values from the simulated plumes were converted into per-pixel plume transmittance spectra with the same LUTs used for the generation of the $\vec{k}$ spectrum, which is an input to the \dx\ retrieval. With this approach of using the same radiative transfer scheme for the forward simulations and for the \dx\ retrieval, we avoid introducing uncontrolled systematic errors in the end-to-end simulation framework (e.g. as from different gas vertical profiles).*"

**Results**

**Figure 2:** Please add a (rough) scale to the image.

A scale bar has been added,

**L182ff / Figure 4:** I really would like to see the difference between proxy and matched filter (as in Fig. 5). Do you find systematic differences between the methods, in particular inside the plume, what might be the reason, and how would they affect your emission estimate?

Fig. 4 has been updated to show the difference map, and these lines have been added to the main text "*Two small clusters of pixels with systematic offsets can be seen in the difference map, at pixel coordinates (10, 60) and (10, 40) corresponding to the larger plume in the subset. However, these enhancements are close to the noise level and have a different sign, leading to an almost zero offset when aggregated to calculate the IME and, subsequently, $Q$.*"

**L198ff:** (see my previous comment on the end-to-end simulator)

Please, see reply to **L143ff.**

**L207ff:** Please provide more information how the DI method has been implemented in this study. Following Chulakadabba et al. (2023), the DI method sums over all pixels along rectangular for difference from the source location to the edge of the detectable plume (except for subtracting the background, which would be about zero for the matched filter). This isn't much different from the IME method, which sums over all pixels in the detectable plume, while excluding the background (which is close to zero).

The main difference between IME and DI method seems to be the effective wind speed. What wind speed is used for the DI method and how does compare to effective wind speed used with the IME method? Would the differences vanish if the same data source (GEOS-FP or HRRR) for the wind speed is used for both methods?

Please, see reply to **L139f** regarding the description of the DI method.

We fully agree with the reviewer in that variations in wind speed data will proportionally affect Q estimates because of the linear relationship between U10 and Q in the IME model (Eq. 2). In this study, our intention was not to evaluate the IME or DI formulations per-se, but to show the validity of our basic IME+GEOS-FP implementation for the estimation of flux rates from the large dataset of plumes detected from the matched-filter output. The DI model constrained with HRRR winds offers an accurate Q estimation framework (although less practical for the processing of large datasets) that we took as a reference for the evaluation of our IME-based Q estimates.

This is discussed in the text as *"each method was constrained with different wind data: the IME-based method is run with GEOS-FP data, as this is the configuration that we apply for the processing of the large plume datasets derived in this work, whereas the DI method is constrained with HRRR wind data, as this is the configuration that potentially provides the most accurate reference for intercomparison with the IME approach"*

**L223f:** It would interesting do discuss the potential for systematic errors inside the plume here (see my previous comment).

Please, see reply to **L182ff / Figure 4.**

**L226f:** I do not see why the detection limit should not always increase with wind speed. However, detection limit can depend on other factors such as turbulence.

Clarification added as: *"which implies that the probability of plume detection is not always inversely proportional to wind speed, but in some cases there is an optimal wind speed for plume detection: low-to-moderate winds enabling the development of a plume covering several pixels with \dx\ values above the noise level."*

**L238f:** What is the uncertainty of the emission rates from the controlled releases?

The caption of Fig. 8 has been updated as *"The metered flux rates correspond to 30-second averages. Error bars in the $y$-axis represent the 1-sigma error for the IME $Q$ estimates from MethaneAIR, and error bars in the $x$-axis represent the standard deviation in the metered flux rate values in the 30-second window."*

**Figure 8:** Why is the RMSE measured in ppb?

Thanks for pointing this out, the legend has been corrected to kg/h (also in Fig. 6)

**L280ff:** Does the wind speed vary spatially and temporally in the campaign area during data acquisition? I expect that would affect the detection limit during the campaign.

*We agree. A clarification has been added as "The GEOS-FP wind product shows average wind speeds of about 3.5\,m/s for RF06, whereas stronger winds of about 5\,m/s are reported in GEOS-FP during the MX025 flights, with a standard deviation of 0.5\,m/s in both cases. The stronger winds may have led to higher detection limits for the MX025 campaign. We have not analyzed spatial and temporal variations of wind speed during data acquisition for each campaign in depth, but such changes would also have an impact on plume detections in each campaign"*

**Figure 12:** Do you see a dependency of flux rates on wind speed?

*We would need a more careful analysis, but from a simple representation of the data included in Fig. 12, we do not see a depency of the flux rate estimates on u10 (see figure below).*

[Figure]

**L292:** Does the albedo change between the campaigns?

*This line has been rephrased as "As mentioned earlier in this work, the detection of a plume in a \dx\ map depends on several factors, including the wind speed, the retrieval noise (driven by at-sensor radiance and local variability in the surface albedo), and the modification of \dx\ gradients by neighboring sources."*

**L292f:** It is unclear what do you mean with "enhanced spatial variability of $\Delta XCH4$".

*The "enhanced spatial variability of $\Delta XCH4$" has been rephrased as "the modification of \dx\ gradients by neighboring sources"*

**Conclusions**

**L312f:** Please specify why a computationally efficient retrieval would improve the detection limit.

*Sentence rephrased as "Our goal was to implement a \dx\ retrieval which was both computationally-efficient and able to maximize the probability of plume detection"*

**L332:** I suggest adding that a major advantage is minimizing the number of false positives.

*Statement added as ", with a minimum rate of false positives"*

---

## Author Response (AR2)

**Response to reviewers (egusphere-2024-3577)**

Dear authors,

The reviewers found the manuscript to be much improved from the previous version, but they have a few remaining points that they insist to be further looked at.

One point is the albedo correction proposed by Foote et al. (2020), which you argued is less relevant for a higher spectral resolution instrument like MethaneAIR, but the reviewer argues that it is nevertheless necessary. I invite you to have a closer look at this point.

The other reviewer would like to see, for completeness and transparency, also the results from the 29 October release experiment, even if conditions were less favourable on that day.

The final critical point concerns the implications of the lower detection limit offered by the matched filter algorithms on the total point source emissions.

For more details, please have a look at the reviewer's comments.

Best regards
Dominik Brunner

Dear Editor,

Please, find enclosed our point-by-point responses to the reviewer's comments. The comments from the reviewers are in black, and our responses are in blue.

Based on the reviewers' comments, the main changes to the manuscript after this revision are:

- Figure 8 has been extended by 2 more panels in order to accommodate the results from the 29 October controlled release campaign, as requested by Reviewer 1.

- A discussion of the impact of plume detection limits on total point source quantifications has been added, as suggested by Reviewer 1.

- An analysis of the effect the albedo correction suggested by Reviewer 2 has been carried out. Some results are shown in this response letter, and a discussion of the convenience of this correction for the processing of MethaneAIR data has been added to the manuscript.

We hope that the current version of the manuscript satisfies the expectations of the Editor and the reviewers.

Sincerely,

Luis Guanter, on behalf of the authors

Report 1

I appreciate the authors' responses to my comments and feel the manuscript has greatly improved. I also appreciate the inclusion of the data with the SI. I still have a few lingering comments before I can recommend for publication.

1. Responding to my original comment #3 - in an effort to be transparent, it is still important to show the controlled release results from 29 October. What do you mean by more stable winds? How are you sure that the conditions in the Permian were of sufficient stability to quantify emissions? It's useful to see controlled release results in a less ideal environment because it gives a general sense of how accuracy depends on environmental conditions out of one's control.

We thank the reviewer for emphasizing the importance of transparency regarding the 29 October controlled release (RF03E).

By "more stable winds," we refer to the fact that wind conditions on 25 October  (RF01E) were more consistent and better aligned between different observational sources and model output. To support this, we compared WRF-LES simulations (driven by HRRR), ASOS observations at nearby airports (P08, CGZ, MZJ), HRRR inputs to the inversion, and ground-based measurements at the release site. We analyzed the period from 16:30 to 21:00 UTC and found that the WRF-LES winds and ground-based winds were significantly more correlated on 25 October (r = 0.41) than on 29 October (r = 0.028). Visual inspection confirmed that 29 October featured more variable and less coherent wind fields. This situation is summarized in these plots:

[Figure]

In any case, in order to satisfy the reviewer's concern, we have now included the results from the 29-October campaign, which has implied to download and processing of a substantial volume of L1B data. The results are shown in the new Fig. 8 (reproduced below), which adds the scatter plot between estimated and in situ flux rates for 29 October, and a comparison of the GEOS-FP and in situ winds for the two dates. We find a bias of about 40% for the 29-Oct data set, which can be largely explain by the substantial overestimation of wind speed by GEOS-FP with respect to the in situ measurements.

This is discussed in the text as *"However, we find an important overestimation of about 40\% in the MethaneAIR flux rate estimates from 29 October, which we attribute to the large overestimation of wind speed that we find in GEOS-FP with respect to the metered wind speeds for that date (Fig.\,\ ref{fig:cr_scatter}). This bad performance of GEOS-FP winds for 29 October is consistent with the poor performance of other wind sources and WRF-LES simulations to reproduce in situ winds for that date."*

[Figure]

2. Regarding my original comment #5 on point vs area sources. Appreciate the response, and I suppose it could be argued that it is out of scope to talk area fluxes in this manuscript, though I generally disagree. That said, putting area fluxes aside, I do believe the newly proposed text is misleading. A central focus of this manuscript is that by application of a matched filter algorithm, the effective detection limit of MethaneAir has reduced. However, in the authors' response, they filter out all the new plumes they detected with this new algorithm for their comparison with the CO2_proxy algorithm - i.e., after filtering out plumes > 200 kg/h, then summing their plume dataset, they get a similar emission total. This greatly undermines what makes this study so compelling!

What I would like to see is one additional sentence or two where the authors describe that when summing all plumes detected from the CO2 proxy approach, they get 26.7 t/h, but when they sum all plumes detected with the matched filter approach, they get 35.9 t/h (I'm just summing the provided RF06 list of detections). This means that there are 9.2 t/h more emissions attributed to point sources that is purely the result of an algorithm update.

This is a fundamentally important point as observing systems are not just the result of instrument specifications but are a result of the algorithms applied to them, and for point source detection, the plume detection protocols as well. And depending on algorithm, you may come to slightly different conclusions about an area/process/etc. The authors should do more highlight this point, in my opinion.

We appreciate this point by the reviewer. The following discussion has been added to the manuscript: "*On the other hand, when using all the detected plumes in our quantification of total emissions (i.e. without filtering out plumes $<$200\,kg/h), we obtain an increase of the total emission estimate of about 9 t/h (RF06) and 6 t/h (MX025) with respect to the Level-4 product. This result confirms the sensitivity of total emission estimates from single plumes to the detection limits offered by the instrument and the processing chain, and suggests that smaller plumes contribute substantially to the totals despite the typical heavy-tailed distribution of point sources \citep[e.g.][] {Cusworth_pnas_2022}*".

Report 2

The authors have addressed most of the reviewers' comments. However, I am not fully convinced by the answers regarding the albedo correction and therefore suggest a minor revision addressing this.

The correction should be necessary because the target spectrum t (=mu k) is calculated from the unit absorption spectrum (k) and the background spectrum (L0), i.e. the radiance spectrum without CH4 enhancement (alpha = 0), which is only approximated by the column-wise mean spectrum (mu). Since the true background spectrum is higher/lower for higher/lower true albedo, CH4 enhancements can be under- or overestimated. Therefore, it is necessary to apply the albedo correction.

Since the correction term will be largest for low/high methane enhancements, i.e. XCH4 > 1000 ppb as seen in the plumes, I would not expect to see the effect in Figure 3, which shows the variability of the background. Even if the effect is small, due to the high spectral resolution of MethaneAir or due to the low variability of the albedo in the flight lines, I think it is absolutely necessary to apply the albedo correction to show its impact on the CH4 enhancements in the plumes and the resulting impact on emission quantification.

Thanks to the reviewer for this point. After a careful evaluation and some tests with real data, we now agree with the reviewer that the albedo correction proposed by Foote et al. would indeed be beneficial for the processing of MethaneAIR data as well. We are including one example below these lines, showing a change of ~4% in the largest plume of Fig. 1. On a plume by plume basis, this number will depend on the similarity between the surface under a given plume and the average spectral albedo of the column, so this effect would manifest itself as an increase in the variance of the Q estimates, rather than as a general offset of the distribution. For this reason, we have opted for including this correction in our processing chain, but not to reprocess the whole MethaneAIR data set presented in this manuscript. This would require an enormous amount of time without a significant impact on the conclusions of the study.

This issue is now discussed in the text as *"We acknowledge that this per-column $\vec{\mu}$ formulation neglects the impact of difference between each pixel's spectral albedo and $\vec{\mu}$. This issue may be alleviated by the albedo correction proposed by \citet{foote}, which adds an ``albedo factor'' to Eq.\,\ref{eq:mf_retr} in order to quantify the difference between $\vec{\mu}$ and $\vec{x}$ for each pixel. The magnitude of this correction will depend on the spatial heterogeneity of the scene. Preliminary tests show that this correction can modify the single \dx\ retrievals by up to 10\% in the Permian (results not shown). However, the sign of the correction can be either positive or negative depending on the albedo of the surface (or surfaces) crossed by a particular plume. For this reason, we can expect that an uncorrected albedo effect may lead to an increase in the scatter of the estimated flux rates within a distribution, but not to a change of the total or the average flux rate of the distribution. In any case, we will implement this correction in future version of the retrieval."*